# “Hands-On” and “Hands-Off” Physiotherapy Treatments in Fibromyalgia Patients: A Systematic Review and Meta-Analysis

**DOI:** 10.3390/biomedicines12102412

**Published:** 2024-10-21

**Authors:** Riccardo Buraschi, Giorgia Ranica, Jorge Hugo Villafañe, Rosa Pullara, Massimiliano Gobbo, Joel Pollet

**Affiliations:** 1IRCCS Fondazione Don Carlo Gnocchi, 20148 Milan, Italy; granica@dongnocchi.it (G.R.); rpullara@dongnocchi.it (R.P.); massimiliano.gobbo@unibs.it (M.G.); jpollet@dongnocchi.it (J.P.); 2Department of Physiotherapy, Faculty of Sport Sciences, Universidad Europea de Madrid, 28670 Villaviciosa de Odón, Spain; mail@villafane.it; 3Musculoskeletal Pain and Motor Control Research Group, Faculty of Sport Sciences, Universidad Europea de Madrid, 28670 Villaviciosa de Odón, Spain; 4Department of Clinical and Experimental Sciences, University of Brescia, 25121 Brescia, Italy

**Keywords:** fibromyalgia, physical therapy modalities, rehabilitation

## Abstract

**Background**: Physiotherapy plays a key role in managing fibromyalgia, a multifaceted disorder, through a combination of active and passive treatments. The purpose of this review is to compare the efficacy of “hands-off” treatments alone versus the combination of “hands-off” and “hands-on” therapies. **Methods**: MEDLINE (PubMed), CENTRAL, and Embase were searched. English-language randomized controlled trials involving adults with fibromyalgia were included. The included studies were divided into subgroups to reduce the possible heterogeneity. We calculated the standardized mean difference or mean difference with 95% confidence intervals for the continuous data according to the outcome measures. We used the risk ratio for dichotomous data of the drop-out rate of the studies. **Results:** We included and analyzed seven RCTs. The meta-analysis showed no significant results in the outcomes, pain, QoL, health status, and drop-out rate. We found significant results (*p* < 0.001) in favor of combining “hands-off” and “hands-on” treatments for the rest quality (SMD 0.72, 95% CI 0.35 to 1.09). **Conclusions**: This review increases the treatment options available for clinicians. Up to now, the main guidelines on managing fibromyalgia suggest only approaches based on “hands-off” treatments. These findings suggest that other approaches based on mixed interventions combining “hands-off” and “hands-on” treatments did not reduce the patient outcomes. Moreover, the mixed intervention led to better results for the patients’ sleep quality than the “hands-off” treatments alone.

## 1. Introduction

Fibromyalgia is a rheumatic syndrome without an underlying organic root. It has specific demographic and social characteristics [1], with a heterogeneous prevalence worldwide from a minimum of 0.2% in Venezuela to a maximum of 6.4% in the United States. Fibromyalgia has higher prevalence values in the female population, from 2.4% to 6.8% [2]. At present, the classification criteria developed by the American College of Rheumatology are merely clinical, and no gold standard tests to identify fibromyalgia are available [3]. Although researchers have introduced various hypotheses related to nervous system dysfunctions and genetic predisposition, fibromyalgia pathogenesis is currently unclear. The unified model of fibromyalgia pathogenesis, introduced by Russel and Larson [4] in 2009, could explain the variety and diversity of its symptoms. These uncertainties about fibromyalgia are associated with its delayed diagnosis. It takes approximately 72 months to arrive at a diagnosis in Israel [5], 42 months in Latin America, and 31 months in Europe [6]. This long path leads the affected people to a lower health status with both psychological disorders and physical impairments. Moreover, patients are referred to different kinds of pharmacological, psychological, and physiotherapy treatments. Patients experience pain as the main symptom and describe it as chronic, widespread, deep, and gnawing. Other symptoms are physical and mental fatigue, stiffness, psychiatric and psychological conditions (e.g., anxiety, depression, and post-traumatic stress disorder), cognitive dysfunction, sleep problems, and autonomic disturbances, and it is also characterized by central sensitization [7,8]. The combination of pain, fatigue, and cognitive difficulties significantly impairs the ability to perform daily tasks, maintain employment, or engage in social activities. Moreover, patients may experience fluctuations in the symptoms’ severity that complicate their ability to plan or commit to activities. The literature strongly recommends physiotherapy to reduce and manage all these symptoms; specifically, exercise therapy and physical activity are the most effective interventions for fibromyalgia patients [9,10]. Moreover, the EULAR (European Alliance of Association for Rheumatology) revised recommendations of 2018 suggest active exercise as the best therapy for fibromyalgia patients [11]. Despite this evidence, other studies have proved the efficacy of passive therapies such as massage [12] and physical agent therapies (e.g., electromagnetic field therapy and laser therapy) [13,14]. The presence of different approaches, both active and passive, is challenging for fibromyalgia patients and the clinicians in charge of them, with a likely risk of wasting resources and time [15]. From this perspective, it is crucial to define which is the best treatment modality. In more detail, there is a compelling need to clarify whether active treatments, also called “hands-off”, are more effective as a standalone intervention or, instead, whether the patient has more benefit from the combination of “hands-off” and “hands-on” therapies (i.e., passive treatments in which patients do not actively contribute to the therapy). We conducted a systematic review to investigate which treatment modality had the best efficacy on pain, quality of life, health status, muscle strength, drug consumption, rest quality, strength, and therapy adherence.

## 2. Materials and Methods

This systematic review was conducted following the PRISMA guidelines [16] and Cochrane Handbook for Systematic Reviews of Intervention (version 6.3) [17]; the manuscript was written fulfilling the PRISMA checklist requirements (Appendix A). The protocol of this review was registered in PROSPERO: CRD42022315218. (https://www.crd.york.ac.uk/prospero/display_record.php?RecordID=315218, accessed on 16 October 2024).

### 2.1. Types of Studies

Only randomized controlled trials (RCTs) were included. We excluded pilot, cross-over, and preliminary trials, protocols, conference and meeting abstracts, and any study designs other than RCTs. Only English studies were assessed for eligibility due to the shortage of funds for the translation process.

### 2.2. Population

We included studies involving adult patients of either sex (>18 years old) with a diagnosis of fibromyalgia.

### 2.3. Types of Interventions and Comparators

In the literature, the terms “hands-off” and “hands-on” treatments are used without a clear and consistent definition. Different authors have varying interpretations of these terms. For example, Hidalgo [18] defines “hands-on” treatments as passive techniques such as spinal mobilization, manipulation, neurodynamic mobilization, and muscle energy technique while considering “hands-off” treatments active techniques like active exercises including core stability. On the other hand, other authors [19,20] only categorize manual therapy as a “hands-on” treatment. Geri et al. [21] consider education and exercise as “hands-off” treatments, and manual therapy and passive treatments are categorized as “hands-on” techniques. The existence of several different acceptations for those terms highlights the need for univocal definitions. For the scope of this review, we decided to define them based on the existing literature and our clinical expertise as follows. During “hands-off” treatments, patients actively perform the assigned tasks, exercises, and activities with verbal and eventual haptic help or guidance. The therapist’s hands are not in contact with the patient unless they serve as a guide or for assistance in performing active exercises. Education, biofeedback, and aquatic exercises are also considered “hands-off” techniques. Instead, in “hands-on” treatments, patients do not actively contribute to the therapy but passively receive it. The therapy is given with hands (e.g., manual therapy and passive mobilizations) or tools and devices (e.g., physical agent therapies or whirlpool therapies) in direct contact with the patient. We excluded the articles in which the type of intervention or control was not sufficiently and exhaustively described to be able to categorize them as “hands-on” or “hands-off” treatments. The RCTs that included treatments based on the combination of “hands-off” and “hands-on” in all study arms were excluded. Studies that compared “hands-off” and “hands-on” treatments alone without the combined interventions were excluded.

### 2.4. Outcome Measures

The primary outcomes considered were pain, quality of life, and the patient’s health status. The secondary outcomes were drug consumption, rest quality, muscle strength, and therapy adherence (measured as the number of drop-outs). Where multiple measures were present for a single outcome, we selected the outcome measure according to the list in the protocol registered in PROSPERO.

### 2.5. Search Strategy

An information specialist designed the search strategy. Then two authors (JP and GR) conducted the electronic database research on 8 February 2024 by consulting MEDLINE (PubMed), CENTRAL, and Embase (Appendix A).

### 2.6. Other Sources

The references of the included records were screened to search for additional articles of interest. Moreover, we searched the protocols published in clinicaltrials.gov and other international clinical trial registers and then contacted the authors of the protocols to check for any published trials not retrieved in the search strategy.

### 2.7. Selection of Studies

All the retrieved articles were imported into Rayyan Software (new.rayyan.ai) [22], and duplicates were automatically removed. Then, two reviewers (JP and GR) independently screened the papers for eligibility by title, abstract, and full-text reading according to the selection criteria. Disagreements between the two reviewers were solved by discussion, and a third author (RB) was consulted in cases of persistent conflict.

### 2.8. Data Extraction

Two reviewers (GR and JP) extracted the data from an Excel spreadsheet. The same reviewers entered the data in RevMan v. 5.4 for the statistical analysis. The following data were extracted: study general information (author, year, title, DOI, journal, country), population (diagnostic criteria, age, gender, and the number of participants enrolled for both experimental and control groups), intervention (the description, frequency, and duration of the experimental and control interventions), and outcome (the outcome assessed and its measure were extracted).

### 2.9. Risk of Bias Assessment

“Cochrane Risk of BIAS tool 1.0” [23] was used to estimate the risk of bias per outcome: random sequence generation (selection bias), allocation concealment (selection bias), the blinding of participants and providers (performance bias), the blinding of outcome assessors (detection bias), incomplete outcome data (attrition bias), and selective reporting (reporting bias). Each class was judged by the high, low, and unclear risk of bias. Two reviewers (GR and JP) independently assessed the risk of bias. A third reviewer (RB) solved the disagreements when necessary. The Robvis R package (https://mcguinlu.shinyapps.io/robvis/) [24] was utilized to create “traffic light” plots.

### 2.10. Data Synthesis

In the process of the data synthesis and meta-analysis, the study employed the calculation of standardized mean differences (SMD) or mean differences (MD) accompanied by 95% confidence intervals (CI) for the analysis of the continuous variables. Owing to the presence of uncontrolled sources of heterogeneity, a random effects model was deemed appropriate for the meta-analysis process. The selection between the MD and SMD was contingent upon the uniformity of the outcome measures; the MD was utilized for identical measures, whereas the SMD was applied for disparate ones. For dichotomous data on the study drop-out rates, the risk ratio (RR) was the metric of choice. To attenuate the level of heterogeneity, the creation of distinct subgroups was proposed, stratified according to the nature of the intervention.

### 2.11. Certainty of Evidence

To detect the certainty of evidence of the primary outcomes considered (i.e., pain, quality of life, and health status), the ‘GRADE handbook for grading the quality of evidence and strength of recommendations’ [25] and GRADEpro GDT Software (McMaster University and Evidence Prime, 2022, https://www.gradepro.org/) were used.

### 2.12. Dealing with Missing Data

Where data were not extractable or missing, we contacted the corresponding author or the first author of the studies. None of the contacted authors replied.

## 3. Results

This systematic review included all the results of the research conducted on 8 February 2024. We found a total of 4561 records. At the end of the screening process, seven studies were included and meta-analyzed (Figure 1). The studies excluded in the full text screening phase are available in Appendix A. The studies included considered 272 patients, all females. Three different diagnosis criteria were used for the patients’ inclusion: the American College of Rheumatology criteria (ACR) 1990, the ACR 2010, and the presence of at least 11 out of 18 tender points (Table 1).

The included studies were Kutlu 2020 [26], Matsutani 2007 [27], Mutlu 2013 [28], Panton 2009 [29], Toprak 2017 [30], Toprak 2020 [31], and Varallo 2022 [32]. The studies were conducted in Turkey [26,27,30,31], Brazil [27], the USA [29], and Italy [32]; they were published between 2009 and 2022. Only one study, Mutlu [28], specified the time since diagnosis, which was more than one year. Moreover, Varallo 2022 [32] included in its trial only fibromyalgia patients with obesity.

### 3.1. Risk of Bias

The risk of bias in all the included studies presented outcome measures that could be affected by group allocation awareness (selection bias). In particular, the blinding of participants and personnel was at high risk of bias in all the studies included, as for the obvious impossibility of blinding an active physiotherapy intervention. However, five studies had a blinded assessor, one study did not report if the assessor was blind or not, and one study had an unblind assessor. Four studies correctly randomized the study participants, while three studies did not sufficiently describe the sequence generation procedure. The allocation of the study participants was not sufficiently described in most of the included studies (i.e., five out of seven studies included). The attrition bias was high in five studies, unclear in one, and low in another one. Only one study provided a clinical trial protocol registration and had a low risk of reporting bias, and the other six studies presented an unclear risk. The presence of other biases was unclear in all the studies included. The graphical representations of the study biases are presented in Figure 2, Figure 3 and Figure 4.

### 3.2. Interventions and Comparators

The full list of the interventions provided in the included studies is reported in Table 1. The included trials were divided into three subgroups to reduce the heterogeneity. The first subgroup was composed of Matsutani 2007 [27], Mutlu 2013 [28], Kutlu 2020 [26], and Varallo 2022 [32], as they all combined a “hands-off” treatment with a “hands-on” treatment based on physical agent therapies. In particular, Matsuani 2007 [27] proposed an intervention based on the active stretching of different muscle groups, comparing it to the same stretching with the addition of laser therapy at the tender points (for this review considered “hands-on” therapy). Mutulu 2013 [28] administered an exercise protocol including cycling and stretching to one group of participants, to which he added TENS for the other group of participants. Kutulu 2020 [26] designed two interventions: one of exercises only and the other of exercises with the addition of vagus nerve stimulation via TENS. Finally, Varallo 2022 [32] proposed exercises based on aerobic training, postural control, and stretching for one group of participants, adding whole body cryotherapy to these exercises for the other study group.

The second subgroup was composed of the studies that combined a “hands-off” treatment with a “hands-on” treatment based on manual therapy (Panton 2009 [29] and Toprak 2017 [30]). Panton 2009 combined resistance training, common to both study groups, with a “hands-on” treatment consisting of ischemic compressions and chiropractic adjustments. Meanwhile, Toprak 2017 [30] proposed the addition of connective tissue massage to a “hands-off” exercise program.

The last subgroup included only Toprak 2020 [31], which combined a “hands-off” stabilization exercises-based treatment with an intervention based on Kinesio taping with the application of three Y-shaped Kinesio tapes at the level of the posterior trunk (on the shoulder blades and thoracolumbar fascia).

**Table 1 biomedicines-12-02412-t001:** Characteristics of the included studies.

Author Year	Diagnostic Criteria	“Hands-Off”				“Hands-Off” + “Hands-On”				Total Duration
		Sample Size (Mean Age ± SD) (Mean (Min–Max))	Intervention	Duration, Frequency, n° of Sessions	Time of Assessment	Sample Size (Mean Age ± SD) (Mean (Min–Max))	Comparator	Duration, Frequency, n° of Sessions	Time of Assessment	
Matsutani 2007 [27]	1990 ACR	10 (45 (31–57))	- Active stretching: stretching exercises, to stretch the scalenes, minor pectoralis, intercostals, diaphragm, paraspinal, hamstring, glutei, triceps suralis, iliopsoas, adductors, internal rotators of the hip, trapezius, deltoid, elbow, fist and finger flexors, subscapular, major pectoralis, and coracobrachialis muscles.	- 60 min twice a week, 10 sessions.	Post treatment (5 weeks)	10 (44 (28–60))	- Active stretching: stretching exercises, to stretch the scalenes, minor pectoralis, intercostals, diaphragm, paraspinal, hamstring, glutei, triceps suralis, iliopsoas, adductors, internal rotators of the hip, trapezius, deltoid, elbow, fist and finger flexors, subscapular, major pectoralis, and coracobrachialis muscles - Laser Therapy (LT): laser was applied at an intensity of 3 J/cm^2^, in continuous mode, with the probe head held at a right angle to the skin at each tender point.	- 60 min twice a week (active stretching). - 60 min twice a week (LT). - 10 sessions	Post treatment (5 weeks)	5 weeks
Panton 2009 [29]	Presence of at least 11 of 18 tender points	10 (50 ± 7)	- Resistance training: patients trained using 9 resistance machines that included the chest press, leg extension, leg curl, leg press, arm curl, seated dip, overhead press, seated row, and abdominal crunch.	- Duration not specified, twice a week, 32 sessions	Post treatment (16 weeks)	11 (47 ± 12)	- Resistance training: patients trained using 9 resistance machines that included the chest press, leg extension, leg curl, leg press, arm curl, seated dip, overhead press, seated row, and abdominal crunch. - Chiropractic Treatment (CT): ischemic compression to tender points on the back of the neck and spine and diversified chiropractic spinal adjustments.	- Duration not specified, twice a week (resistance training). - 10 min, twice a week (CT). - 32 sessions	Post treatment (16 weeks)	16 weeks
Mutlu 2013 [28]	1990 ACR	30 (43 ± 11)	- Exercise: warm up, cycling, stretching, and strengthening exercises and cooldown.	- 40 min, three days a week. 36 sessions	Post treatment (12 weeks)	30 (46 ± 9)	- Exercise: warm up, cycling, stretching, and strengthening exercises and cooldown. - TENS: applied to the most painful areas (80 Hz).	- 40 min, three days a week (exercise). - 30 min, five days a week for the first 3 weeks of treatment (TENS). 36 sessions	Post treatment (12 weeks)	12 weeks
Toprak 2017 [30]	1990 ACR	20 (40 ± 10)	- Exercise: warm up, aerobic, and strengthening exercises, cooldown, and stretching including neck, trunk, and upper and lower limb muscles.	- 60 min, twice a week. 12 sessions	Post treatment (6 weeks)	20 (43 ± 8)	- Exercise: warm up, aerobic, and strengthening exercises, cooldown, and stretching including neck, trunk, and upper and lower limb muscles. - Connective Tissue Massage (CTM): applied in the lumbosacral, lower thoracic, scapular, interscapular, and cervical regions.	- 60 min, twice a week (exercise). - 5/20 min, twice a week (CTM). 12 sessions	Post treatment (6 weeks)	6 weeks
Toprak 2020 [31]	1990 ACR	19 (44 ± 10)	- Exercise: postural training, stabilization exercise with contractions of the multifidus and transversus abdominis muscles.	- 60 min, twice a week. 12 sessions	Post treatment (6 weeks)	17 (38 ± 24)	- Exercise: postural training, stabilization exercise with contractions of the multifidus and transversus abdominis muscles. - Kinesio taping (KT): three Y-shaped Kinesio tapes with a width of 5 cm and thickness of 0.5 mm were used for the technique. The base of the first Y-shaped tape was placed approximately ½ to 1 inch below T12 in standing position with no tension. The tails of the Y-shaped tape were applied from the superior edge of the scapula to the axilla. Two other bands were bilaterally applied for the thoracolumbar fascia.	- 60 min, twice a week (exercise). - Duration not specified, twice a week (KT). 12 sessions	Post treatment (6 weeks)	6 weeks
Kutlu 2020 [26]	2010 ACR	25 (39 ± 9)	- Exercise: strengthening, stretching, isometric, and posture exercises, targeting the body and upper and lower extremities.	- Duration not specified, 5 days a week, twice a day. 40 sessions	Post treatment (4 weeks)	27 (39 ± 8)	- Exercise: strengthening, stretching, isometric, and posture exercises, targeting the body and upper and lower extremities. - Vagus Stimulation (VG): carried out with a TENS device, which has specially designed surface electrodes in the shape of earphones, the size of which can be selected according to ear size. Electrodes were placed to correspond with the inner and rear surfaces of the tragus and the concha for both ears.	- Duration not specified, 5 days a week, twice a day (exercise). 40 sessions - 30 min, five days a week (VG). 20 sessions	Post treatment (4 weeks)	4 weeks
Varallo 2022 [32]	ACR criteria version not specified	23 (49 ± 7)	- Exercise: personalized progressive aerobic training, postural control exercises, and progressive strengthening exercises.	- 60 min, two times per day.	Post treatment (2 weeks)	20 (53 ± 8)	- Exercise: personalized progressive aerobic training, postural control exercises, and progressive strengthening exercises. - Whole Body Cryotherapy (WBC): patients were exposed to extremely cold, dry air at −110 °C for 2 min in a cryochamber (Artic, CryoScience, Rome, Italy).	- 60 min, two times per day. 20 sessions - 2 min WBC, 10 sessions	Post treatment (2 weeks)	2 weeks

### 3.3. Effects of Interventions

#### 3.3.1. Primary Outcome (Pain, Quality of Life, Patient’s Health Status)

It was possible to meta-analyze the results of the studies included for the three primary outcomes considered; the certainty of evidence for the three primary outcomes considered was low, downgraded by one level, respectively, for the risk of bias concerns and imprecision. For the pain outcome, we pooled all seven studies together. The meta-analysis showed no significant results both in the subgroups and total analysis (SMD 0.22, 95% CI −0.10 to 0.55). The total heterogeneity was low (I^2^ = 44%) and moderate (I^2^ = 69%) in the subgroup “physical agents therapy” (Figure 5). Five studies considered quality of life as an outcome. For the quality of life, the combination of “hands-off” treatment and Kinesio taping showed a significant result (SMD 0.92, 95% CI 0.23 to 1.61), even if the total result of the meta-analysis showed a non-significance (SMD 0.08, 95% CI −0.32 to 0.49) with a moderate heterogeneity (I^2^ = 52%) (Figure 6). All seven studies assessed health status through the Fibromyalgia Impact Questionnaire. The meta-analysis produced non-significant results (MD 3.53, 95% CI −0.13 to 7.19). The total heterogeneity was low to moderate (I^2^ = 42%) (Figure 7).

#### 3.3.2. Secondary Outcome (Drug Consumption, Rest Quality, Muscle Strength, Therapy Adherence)

In the meta-analysis, it was possible to include the secondary outcome results of the rest quality and the number of drop-outs. Three studies were pooled for the outcome of the rest quality, with significant results (*p* < 0.001) in favor of “hands-on” combined with “hands-off” treatments (SMD 0.72, 95% CI 0.35 to 1.09). The total heterogeneity was low (I^2^ = 0%) (Figure 8). The meta-analysis produced non-significant results for the outcome of the number of drop-outs (RR 1.10, 95% CI 0.57 to 2.12) with a low heterogeneity (I^2^ = 0%) (Figure 9). Only one study [29] considered muscle strength as an outcome, which improved significantly in both groups. Drug consumption was not analyzed in any of the included studies.

## 4. Discussion

This systematic review aimed to compare the efficacy of “hands-off” treatments as a standalone intervention to the combination of “hands-off” and “hands-on” therapies in fibromyalgia management. The results of this review did not show appreciable differences between the two approaches on pain, quality of life, and health status. Only rest quality seemed to be influenced by the two combined treatments.

In the complex management of fibromyalgia, physiotherapy plays a crucial role. Two main types of therapies are predominantly utilized: “hands-off” or active therapy and “hands-on” or passive therapy. Up to now, the only treatment the EULAR guidelines recommend is exercise therapy [11]. This review shows that adding “hands-on” treatments to “hands-off” treatments does not change the outcomes and the adherence to the treatment. From a clinical perspective, the findings of this review increase the clinical options available for physical therapists. Including “hands-on” treatments in the therapeutic plan would be helpful for patients suffering from inadequate sleep quality. This is an interesting result, as subjects with fibromyalgia report an increased prevalence of sleep disturbance and reduced sleep quality [33,34]. Moreover, sleep quality is related to pain perception and intensity [35]; even if this systematic review did not show a significant effect on pain relief after treatment, it would be interesting if future studies address the intermediate and long-term effects on pain relief and sleep quality of the combined treatment. It is interesting to notice that, among the interventions causing an increase in rest quality, there is the Kinesio taping application [31], which has been shown to have little to no effect in different musculoskeletal disorders, according to some studies [36,37,38]. Our review showed no effects on pain relief, but only on rest quality. Instead, cryotherapy, in our results, seems to produce a small reduction in pain and a significant improvement in sleep quality in fibromyalgia patients. According to the available literature, there is moderate evidence of cryotherapy use in acute settings, while for the management of long-term pain and dysfunctions, the certainty of evidence is low [39]. However, the sub-group analysis of the physical agents therapy did not show a significant reduction in pain. A recent review by Cuenca-Martinez et al. showed how strength training may improve rest quality [40]. In our review, only Panton 2009 [29] assessed strength as an outcome but performed only a resistance training program. Four studies administered strength training (i.e., Mutlu 2013 [28], Toprak 2017 [30], Kutlu 2020 [26], and Varallo 2022 [22]) without assessing strength as an outcome. Only two studies [30,32] out of the four performing strength training considered a progression in their program. However, only Toprak 2017 [30] specified the criteria for progression (i.e., a perceived pain or fatigue level lower than severe).

The choice of one of the two treatment options did not affect the drop-out rate, and this is crucial because, even if one of the main reasons for using “hands-on” treatments is to enhance a therapeutic relationship with the patient [21], in the management of fibromyalgia, the use of “hands-off” treatments alone seems not to increase the drop-out rate.

The systematic review of Garijo et al. [41] reported similar results to those retrieved in this review, although some differences exist. Our review, as previously stated, focused on specific interventions and did not cover all the non-pharmacological interventions, while Garijo et al. included many studies with a wide variety of non-pharmacological interventions. Moreover, the authors reported only the effects of the different therapies for each study, with no meta-analysis combining different studies considering the heterogeneity of the available data. For the methodological quality assessment, they used the PEDro scale, which is less reliable than Cochrane RoB 1.0, to assess bias and, indirectly, methodological quality [42]. Instead, Sosa-Reina et al. showed the beneficial effect of exercise therapy, but the efficacy was compared to heterogeneous treatments within the performed meta-analysis (e.g., usual care, relaxation exercises, stretching, etc.) [43]. This approach could have influenced the results. The definition proposed by this review of “hands-off” treatments includes different kinds of treatments. However, the studies included in this review used only exercise therapy in different forms (e.g., aerobic training and strength training) or active stretching. The literature proved the efficacy of other “hands-off” therapies according to our definition, including cognitive behavioral therapies [44], hydrotherapy [45], meditation therapy [46], and mindfulness therapy [47]. Further studies should prove the efficacy of this kind of interventions against the combined interventions.

### 4.1. Clinical Implications

“Hands-off” therapies have been shown to be as effective as a miscellaneous intervention of “hands-off” and “hands-on” therapies. This is an important result as the effect created by the touch of the therapist did not increase the QoL and health status or decrease the pain and drop-out rate; these results agree with the suggestions of the EULAR guidelines. “Hands-off” treatments are less time-consuming, and their effect is not therapist-related; this opens interesting insights into the possible efficacy of telerehabilitation [48] for the management of fibromyalgia.

### 4.2. Limitations

The lack of high-quality studies with appropriate sample sizes and coherent methodology conditions affects these results. Only six trials met the inclusion and exclusion criteria, presenting several concerns about their risk of bias. These concerns did not allow us to conduct a sensitivity analysis. The sample size of the included studies was consequently limited to 229 subjects, all females; this can limit the consistency of our results. On the other hand, this is also the first review that analyzed the differences between “hands-off” and “hands-on” treatments in fibromyalgia patients, clearly defining which treatments need to be included in the “hands-on” or “hands-off” groups.

## 5. Conclusions

This review opens interesting possibilities for managing fibromyalgia. It not only shows that “hands-off” treatments seem to be effective, but that the combination of “hands-off” and “hands-on” treatments shows a similar effect, although the application of this approach should be carefully evaluated according to the available resources. Moreover, the combined treatment provides small but significant improvements in sleep quality, while the adherence (drop-out rate) of the two groups did not produce significant differences. Further studies should consider the comparison of other interventions to increase the knowledge about the different treatment choices.

## Figures and Tables

**Figure 1 biomedicines-12-02412-f001:**
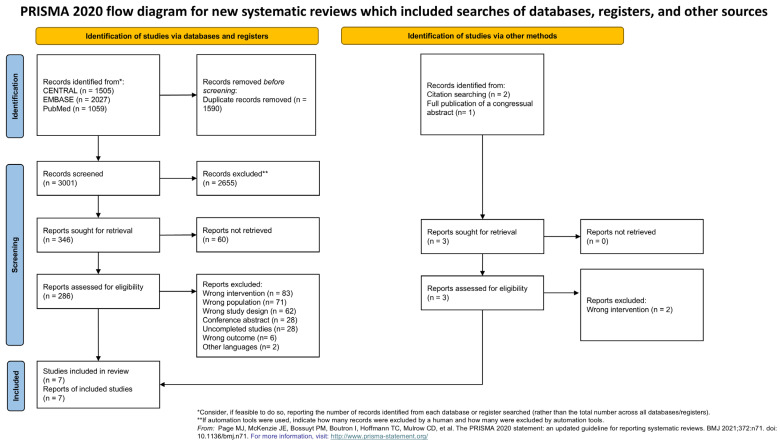
PRISMA flow chart [16].

**Figure 2 biomedicines-12-02412-f002:**
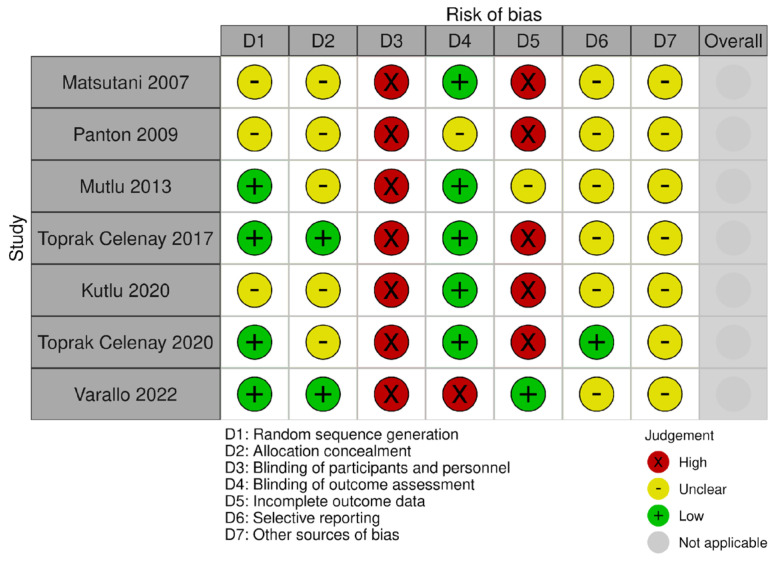
Risk of bias for pain outcome [26,27,28,29,30,31,32].

**Figure 3 biomedicines-12-02412-f003:**
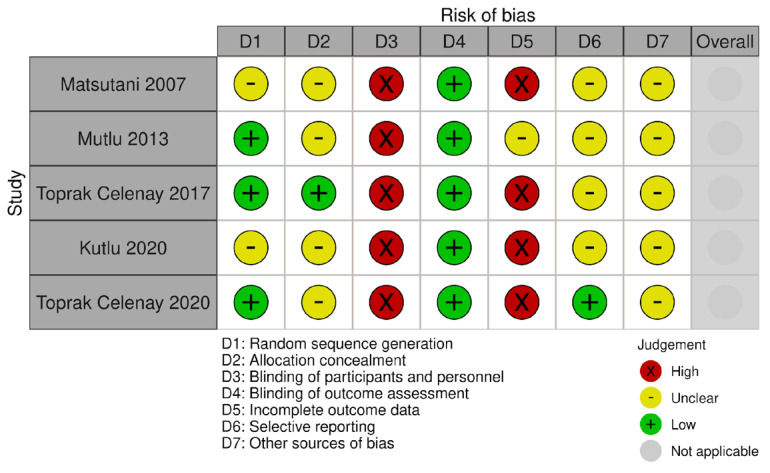
Risk of bias for quality of life outcome [26,27,28,30,31].

**Figure 4 biomedicines-12-02412-f004:**
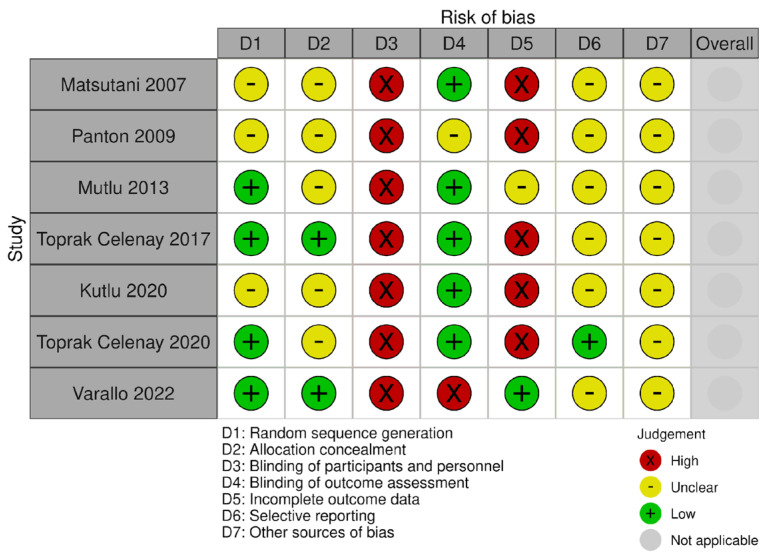
Risk of bias for health status outcome [26,27,28,29,30,31,32].

**Figure 5 biomedicines-12-02412-f005:**
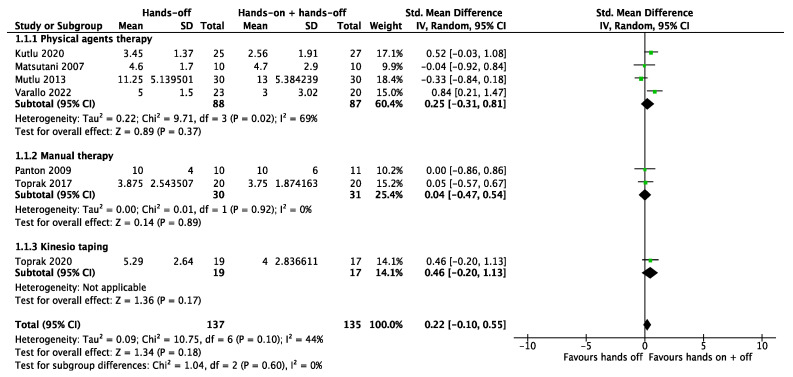
Meta-analysis of studies. Outcome: pain [26,27,28,29,30,31,32].

**Figure 6 biomedicines-12-02412-f006:**
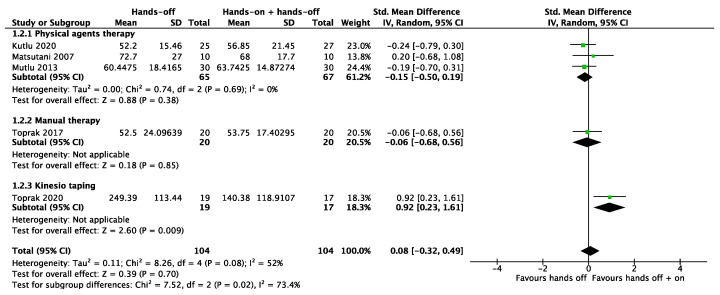
Meta-analysis of studies. Outcome: quality of life [26,27,28,30,31].

**Figure 7 biomedicines-12-02412-f007:**
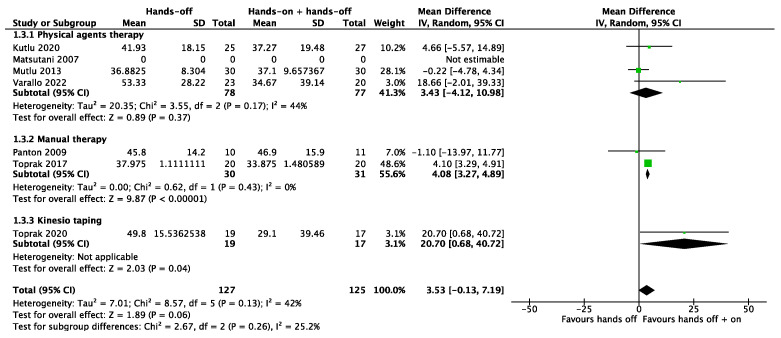
Meta-analysis of studies. Outcome: health status [26,27,28,29,30,31,32].

**Figure 8 biomedicines-12-02412-f008:**
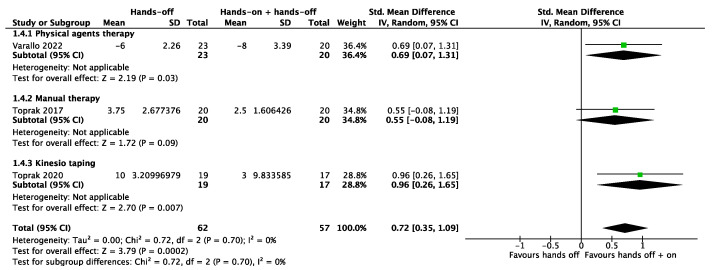
Meta-analysis of studies. Outcome: rest quality [30,31,32].

**Figure 9 biomedicines-12-02412-f009:**
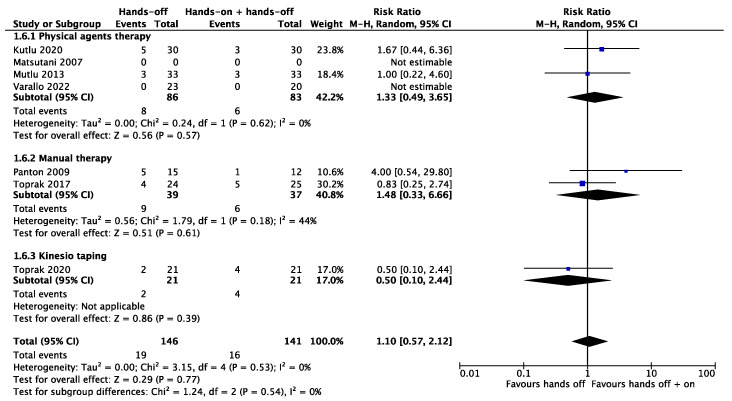
Meta-analysis of studies. Outcome: drop-out rate [26,27,28,29,30,31,32].

## Data Availability

All data are available through a request to the corresponding author.

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
