# Peer review of "“Hands-On” and “Hands-Off” Physiotherapy Treatments in Fibromyalgia Patients: A Systematic Review and Meta-Analysis"

_biomedicines, 2024, doi:10.3390/biomedicines12102412_

Round 1
Reviewer 1 Report
Comments and Suggestions for Authors
After reading the interesting systematic review by Buraschi et al. submitted for peer review to the journal Biomedicines, I can comment on the following:
1. The study was very well done, following a rigorous protocol, from the registration of the study in PROSPERO to the performance of the meta-analysis.
2. This work makes a significant contribution to the field of physiotherapy, because by analyzing the evidence (7 clinical trials) it shows that the physiotherapist's time could possibly be saved by reducing the number of interventions ("hands-on") that are apparently not very effective for the patient.
3. During the review I found some minor problems in the wording of the document. For example: L40 "Until now". It would be better to use "at present". I suggest polishing the document for final release.
4. I suggest using the rovis tool plots (https://www.riskofbias.info/welcome/robvis-visualization-tool) to improve the presentation of the data, considering the two classic figures, light traffic and weighted bar plots
Comments on the Quality of English LanguageMinor problems detected
Author Response
- The study was very well done, following a rigorous protocol, from the registration of the study in PROSPERO to the performance of the meta-analysis.
Thank you for your gratifying comment.
- This work makes a significant contribution to the field of physiotherapy, because by analyzing the evidence (7 clinical trials) it shows that the physiotherapist's time could possibly be saved by reducing the number of interventions ("hands-on") that are apparently not very effective for the patient.
Thank you, we appreciate your comment.
- During the review I found some minor problems in the wording of the document. For example: L40 "Until now". It would be better to use "at present". I suggest polishing the document for final release.
Thank you for the suggestion, we’ve fixed the article accordingly.
- I suggest using the rovis tool plots (https://www.riskofbias.info/welcome/robvis-visualization-tool) to improve the presentation of the data, considering the two classic figures, light traffic and weighted bar plots
Thank you, we improve the presentation of RoB graph using the suggested tool.
Comments on the Quality of English Language: Minor problems detected
We have fixed English quality, thank you for your suggestion.
Reviewer 2 Report
Comments and Suggestions for Authors
Dear Authors,
This SR report is comprehensive and deals with important clinical and research issues. Please find my concerns below.
Minor issues:
introduction. l. 36-37 - the description of the health problem should be more comprehensive, with references (specifically to functioning, not only to the medical diagnosis)
Table 2 - caption 'Table of contents' is vague to me - please provide a specific caption, such as 'characteristics of the included RCTs' or equivalent.
Main comment: there is no supplementary file(s) such as the PRISMA checklist and other, indicated at PRISMA, e.g. list of excluded studies, with reasons. Then table 1 - search strategy, would also go as a supplementary file.
Author Response
- introduction. l. 36-37 - the description of the health problem should be more comprehensive, with references (specifically to functioning, not only to the medical diagnosis)
Thank you for the comment, we described the wide umbrella of symptoms and functioning limitations occurring to patients with fibromyalgia (lines 50-55). However, we appreciate your suggestion, and we had rewritten this part with a better focus on the issue of functioning (lines 58-61)
- Table 2 - caption 'Table of contents' is vague to me - please provide a specific caption, such as 'characteristics of the included RCTs' or equivalent.
Thank you. We’ve changed the table title into “characteristics of the included studies”
- Main comment: there is no supplementary file(s) such as the PRISMA checklist and other, indicated at PRISMA, e.g. list of excluded studies, with reasons. Then table 1 - search strategy, would also go as a supplementary file.
Thank you. We add PRISMA checklist in the supplementary materials, as well as the search strategy table. We also add the list of excluded studies with reasons.